evolution, behaviour, ecology

life-history theory, interdisciplinarity, cultural evolution, bibliometrics, pace of life

**Author for correspondence:**
Daniel Nettle
e-mail: daniel.nettle@ncl.ac.uk

# The evolution of life-history theory: a bibliometric analysis of an interdisciplinary research area

Daniel Nettle[1] and Willem E. Frankenhuis[2]

[1]Centre for Behaviour and Evolution, Institute of Neuroscience, Newcastle University, Newcastle, UK
[2]Behavioural Science Institute, Radboud University, Nijmegen, The Netherlands

DN, 0000-0001-9089-2599; WEF, 0000-0002-4628-1712

The term 'life-history theory' is a familiar label in several disciplines. Life-history theory has its roots in evolutionary models of the fitness consequences of allocating energy to reproduction, growth and self-maintenance across the life course. Increasingly, the term is also used in the conceptual framing of psychological and social-science studies. As a scientific paradigm expands its range, its parts can become conceptually isolated from one another, even to the point that it is no longer held together by a common core of shared ideas. Here, we investigate the literature invoking the term 'life-history theory' using quantitative bibliometric methods based on patterns of shared citation. Results show that the literature up to and including 2010 was relatively coherent: it drew on a shared body of core references and had only weak cluster divisions running along taxonomic lines. The post-2010 literature is more fragmented: it has more marked cluster boundaries, both between the human and non-human literatures, and within the human literature. In particular, two clusters of human research based on the idea of a fast–slow continuum of individual differences are bibliometrically isolated from the rest. We also find some evidence suggesting a decline over time in the incidence of formal modelling. We point out that the human fast–slow continuum literature is conceptually closer to the non-human 'pace-of-life' literature than it is to the formal life-history framework in ecology and evolution.

## 1. Introduction

The term 'life-history theory' has become a familiar one. It is found as a short-hand way of introducing expectations and predictions in papers from ecology and evolutionary biology [1–3], but also psychology [4–7], anthropology [8], public health [9], criminology [10] and even accountancy [11]. Life-history theory has recently been characterized as offering the unifying meta-theory that the social sciences currently lack [12]. Thus, life-history theory is a potentially key inter-disciplinary bridge, connecting diverse human-focused disciplines to evolutionary theory, and hence grounding human-specific knowledge in a general framework that also applies to all other living organisms. However, successful fulfilment of this bridging function relies on the term 'life-history theory' referring to approximately the same body of ideas in all of the diverse areas in which it is used.

As scientific areas grow, they often fission into partially isolated sub-areas. Researchers in a field spontaneously form fluid and informal sub-groups and rely on and credit the ideas generated within their group more than those of surrounding ones. These sub-groups are akin to 'demes' in population biology: partially genetically isolated sub-populations that can form the basis for local adaptation [13]. Demic structure in science is not necessarily a bad thing: once the demes are well enough separated, theories and assumptions can

evolve rapidly within each deme, generating conceptual diversity and adapting to the specific phenomena that deme deals with [13–15]. However, just as in population biology, the existence of demic structure raises classification problems: when are we dealing with locally adapted varieties of the same kind of thing, and when are we giving the same name to what are in fact different kinds of thing?

The emergence of demic structure can be detected in citation patterns, which can be studied quantitatively using bibliometrics. Bibliometrics originated in the early twentieth century [16], with key methodological developments in the 1960s [17–19]. Its more recent development has been facilitated by the availability of comprehensive searchable databases, such as Web of Science (Clarivate Analytics, www.webofscience.com). Two important and related goals of bibliometrics are the construction of maps—that is, representations of which parts of a literature are most closely connected to one another [20]; and the detection of clusters—that is, subsets of a literature that are more closely connected internally than they are to parts outside the cluster [21]. Clusters detected in the literature imply demic structure in the research field. Map-making and cluster analysis have allowed researchers to address questions about the structure of science as a whole [22], as well as of individual disciplines [23] or inter-disciplinary research areas [24].

Scholars typically locate the foundational ideas of life-history theory in formal models of the evolution of life-history traits dating from the mid- and late twentieth century (for influential early sources, see [25,26]). These models offered tools for explaining variation in how organisms should allocate energy to reproduction—or alternatively to other activities such as growth or somatic maintenance—across the life course, and across offspring. The body of these models was classically reviewed in books by Stearns [27] and Roff [28,29], and expanded by the development of state-dependent modelling in the 1980s and 1990s [30,31]. The authors of these works did not typically use the term 'life-history theory', preferring 'life-history evolution' or 'evolution of life histories/life-history traits'; only later did the term 'life-history theory' come to be employed. So dominated was the field of life-history evolution by formal modelling that in 1976, Stephen Stearns was able to complain that the field had too much theory and not enough data [25]. In 1980, he estimated that the proportion of papers on life-history problems based solely on mathematical models was stabilizing at about 30%, which he described as 'a healthy balance' between theory and data [32].

By theory, Stearns [32] meant the practice of formal mathematical modelling of fitness in relation to life-history traits, rather than any particular empirical claim that might arise from such models. However, in psychology and most social sciences, the term 'theory' refers to frameworks that are not formalized and are at least to some extent inductive (based on typical patterns in data) rather than deductive (based on logical inferences from axioms). If 'life-history theory' has adapted to the substrate of psychology and the social sciences as it has spread, we should expect decreasing use of formal models and an increasingly close association of 'life-history theory' with a characteristic empirical pattern. Our impression (from, for example, [4]) is that this has indeed happened. In many psychological and social-science works, 'life-history theory' is used as a near-synonym for the 'fast–slow continuum' (or 'fast' and 'slow' strategies). This is the

descriptive generalization that variation between species or between individuals can be organized onto a principal axis from early maturation and reproduction, small body size, large numbers of offspring and low parental investment, at one end, to late maturation and reproduction, large body size, small numbers of offspring and high parental investment at the other. A full review of the sources, varieties of and evidence for the 'fast–slow continuum' idea is beyond our scope here. We merely hypothesize that it might play a different role in 'life-history theory' in psychology and social science as compared to ecology and evolution.

We used bibliometric tools to investigate the structure of the literature that uses the term 'life-history theory'. We make no claim that this captures or represents the whole of the literature on life-history evolution (though see electronic supplementary material, §1, for more comprehensive searches of the broader literature, which lead to many similar conclusions). Rather, we were interested in the evolution of the use of this particular label. We first sought to document the incidence of the term 'life-history theory' over time. We then examined whether, as it has expanded its range, 'life-history theory' has remained a coherent, connected research programme; or whether it has developed demic divisions. If there are divisions, we aimed to understand where they lie and how deep they are. We investigated the frequency of formal evolutionary modelling in different parts of the literature and also the frequency of appeal to the 'fast–slow' continuum. Different frequencies of these in different parts of the literature may indicate divergence, both methodological and substantive, in what 'life-history theory' is being used to mean.

## 2. Material and methods

Bibliometric analysis requires a choice of nodes and a choice of connections. Nodes can be individual publications, journals, subject categories or authors. Here, we selected individual publications as the nodes, since using life-history theory is a property of an individual work, not of a journal or an author. To identify connections among works, we used bibliographic coupling [33]. Two works are bibliographically coupled if there exists some third work that they both cite; and the more such cited works in common there are, the stronger the coupling. Thus, bibliographic coupling provides a measure of the extent to which different works draw on the same set of intellectual influences. For a set of published works, the matrix of all possible pairwise bibliographic coupling strengths can be used to make a distance-based map that places works that have more similar lists of cited references closer together, and also for the detection of clusters of works whose cited reference lists are particularly similar to one another. Tools for doing this are freely available using the VOSviewer software [34].

We searched Web of Science (Core Collection) on 12 November 2018 for the term 'life-history theory' in 'topic' (for discussion and justification of this search term, see electronic supplementary material, §1). A 'topic' search finds occurrences of the search term in the title, abstract, keywords and keywords plus (keywords plus are automatically generated by Web of Science). There were 1841 records meeting the search criterion. Web of Science searches are not sensitive to hyphenation, and thus our search also returned occurrences of 'life-history theory'. The Web of Science database goes back to 1970. Records were downloaded as text files for bibliometric coupling analysis in VOSviewer [34] and as a BibTeX database for ancillary analysis using R package 'bibliometrix' [35]. The median publication year of papers in the dataset was 2010. Since we were interested in

how the structure of the literature has changed over time, we divided the dataset into two parts: publications appearing up to and including 2010 ($n = 911$), and those appearing later than 2010 ($n = 930$). We also analysed the whole dataset together (electronic supplementary material, §2), and using a finer-grained division into four time periods (electronic supplementary material, §3).

An alternative to bibliographic coupling is co-citation analysis. In co-citation analysis, the nodes on the resulting maps are not the papers found by the literature search, but the papers *cited by* the papers found in the literature search (i.e. those papers that, in bibliographic coupling, determine the proximity). Maps using co-citation analysis were generally similar to those presented here (electronic supplementary material, §4). However, we preferred bibliographic coupling to co-citation for our main analysis, as a number of the nodes in the co-citation maps were works that were not themselves about life-history theory (for example, papers on statistical or measurement methods). Moreover, the bibliographic coupling has previously been argued to produce slightly more accurate maps of a research area [36].

We created maps based on the 500 documents with the greatest link strength. This is recommended by VOSVIEWER authors [37] because it avoids scattered outliers that obscure global patterns. Using the largest connected set of documents instead produces a similar shape for the core of the maps but slightly changes the detected cluster structure. We used the fractional counting method, which equalizes the weight given to each paper [38], including all papers returned by the search regardless of whether they had been cited. We set the cluster resolution parameter at 0.80, with minimum cluster size 20. The choice of cluster resolution is a research judgement, designed to produce an interpretable number of clusters. In the electronic supplementary material, §5, we report the effects of varying the cluster resolution in steps from 0 to 1. To normalize the maps for visualization, we used the fractionalization method (for full details of the options available in VOSVIEWER, see [34]).

Having created the maps, in order to gain some understanding of the content of the identified clusters, we selected a sample of 10 papers that had been assigned to each cluster (see electronic supplementary material, §6, for lists of the sampled papers). We chose papers that either presented empirical data or a formal model and that mentioned life-history theory more than just as a keyword. Among papers meeting these criteria, we selected the 10 with the highest total link strength from each cluster. For the 10 sample papers from each cluster, we downloaded the full electronic version and scored four variables: the taxa of organism from which empirical data are reported; the presence of a formal model; the general topic; and whether they specifically mention the 'fast–slow continuum' or 'fast/slow life-history strategies'. Based on the scores for the sample papers on these variables, we assigned names to the clusters. In addition, we tabulated the 10 references most frequently cited by the papers in each cluster. We also counted how many of these top-cited references presented formal models.

For each time period separately, we calculated indices of connection between papers within and across clusters. Two papers are connected in bibliographic coupling analysis if there is at least one reference that both papers cite. For each possible pair of clusters, including the pairing of a cluster to itself, we calculated the number of possible connections that could exist ($k_i \times k_j$ for two different clusters with $k_i$ and $k_j$ papers in them respectively, and $k_i \times (k_i - 1)$ for connections among papers in the same cluster). We then computed how many of those possible connections actually existed. This gives an index that ranges from 0 (there are no bibliographic connections between the two

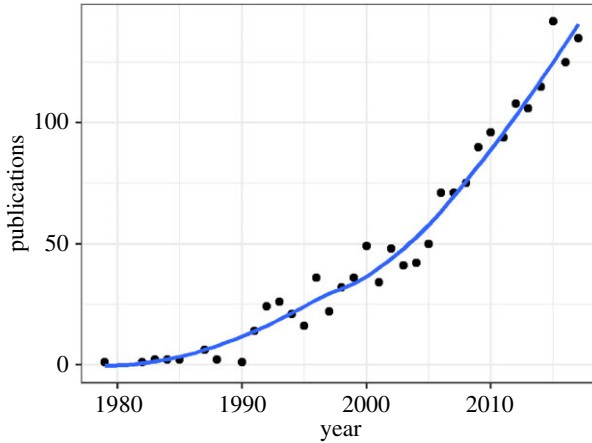

**Figure 1.** Number of publications found using the topic search term 'life-history theory' in Web of Science. The year 2018 is not shown in this figure as it was not complete at time of writing. Line represents a loess (locally estimated non-parametric) fit. (Online version in colour.)

clusters) to 1 (every paper in the first cluster shares a cited reference with every paper in the second). As well as the absolute value of the connection index, we also computed two measures of its inequality. First, we calculated the Gini coefficient of each cluster's connection indices. A connection Gini of 1 for a cluster means that all of its connections are within-cluster and none to other clusters. A connection Gini of 0 means that connections from that cluster are perfectly equally distributed across all clusters (0 is impossible in practice because no clusters would be detected without some inequality in connection). Second, we calculated the within/between connection ratio. This is the ratio of a cluster's within-cluster connection index to the mean of its connection indices to all the other clusters.

## 3. Results

The earliest reference to 'life-history theory' identified by the search was from 1979 [39]. Use of the term was rare and sporadic until 1990, since when there has been steady growth in the number of papers using the term (figure 1). From 2012 onwards, there have been more than 100 new papers every year.

### (a) Structure of the earlier literature (up to and including 2010)

The structure of the earlier literature was radial, with short arms protruding from a central core (figure 2a). There were five clusters. The primary differences between the clusters appeared to be taxonomic (see electronic supplementary material, §7 and table S1). In our sample papers, cluster A1 (red) featured birds exclusively; A4 (yellow) featured reptiles in 7/10 sampled papers and A5 (purple) featured non-human mammals in 9/10 sampled papers. Cluster A3 (blue) was a mixture of work on invertebrate animals, fish and amphibians. Cluster A2 (green) captured all of the research on humans. Some non-human primate data also appeared in the sample papers from cluster A2, often in comparison to humans. Unlike the taxa, the topics of the study did not obviously differ strongly by cluster, with fundamental life-history questions concerning reproductive effort and its costs, parental investment and growth, appearing in all clusters. Formal models were found in 16 of the 50 papers

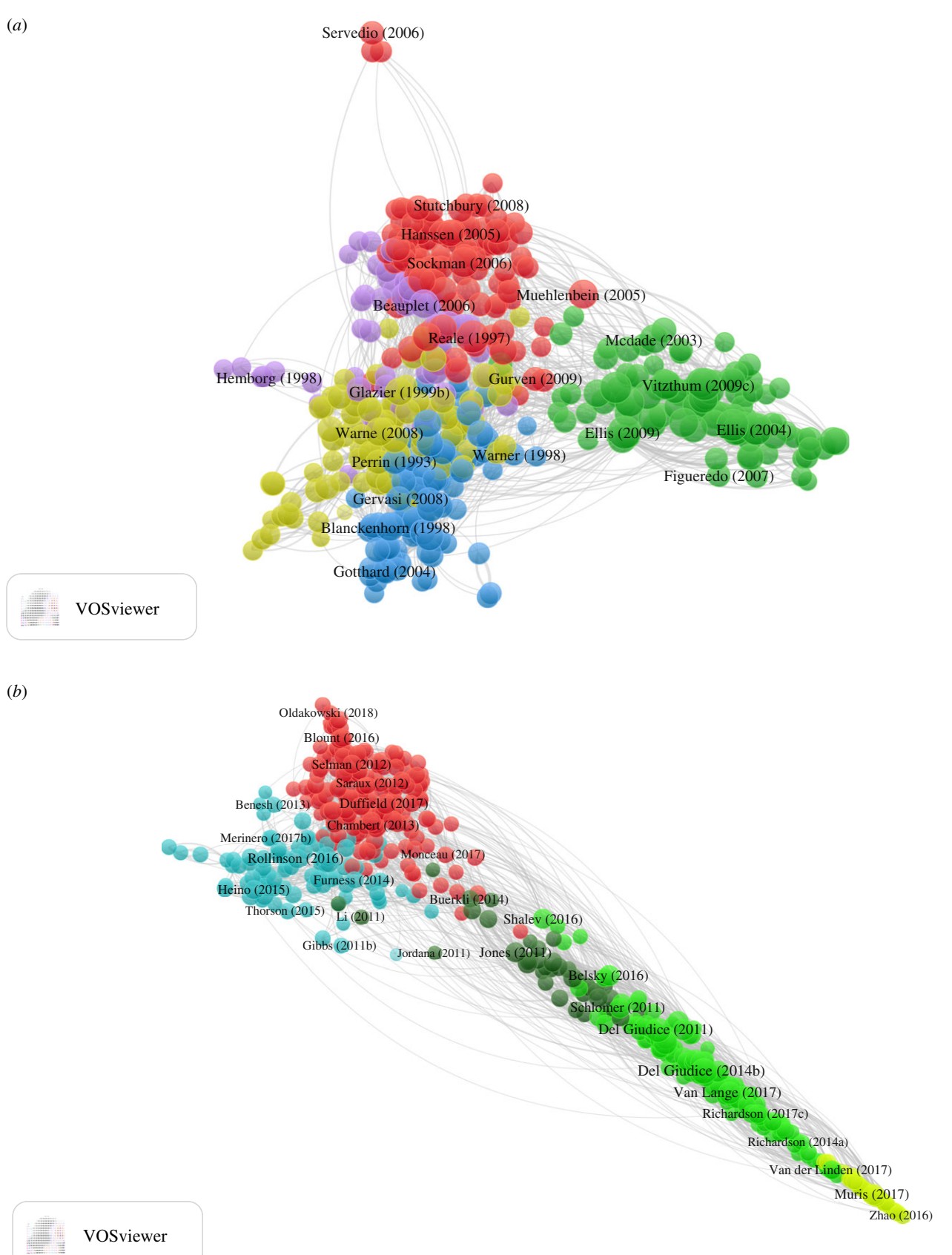

**Figure 2.** Maps of the life-history theory literature based on bibliographic coupling. (a) The literature up to and including 2010. (b) The literature published after 2010. For details of parameter values used, see material and methods. For interactive online versions with links to the papers displayed, see tiny.cc/LHTearlymap and tiny.cc/LHTrecentmap (requires Java, may involve downloading and opening a file). (Online version in colour.)

sampled in this period, distributed across all clusters except A5. The lists of 10 top-cited references for the five clusters of this time period showed some overlap, with Stearns [27] appearing in the top 10 for every cluster (electronic supplementary material, table S1). For all clusters, three or more of the top 10 cited references were sources that presented formal evolutionary models. In this earlier time period, the idea of the 'fast–slow continuum' or 'fast/slow life-history strategies' occurred in four of the 50 sample papers (one in cluster A2, three in A5).

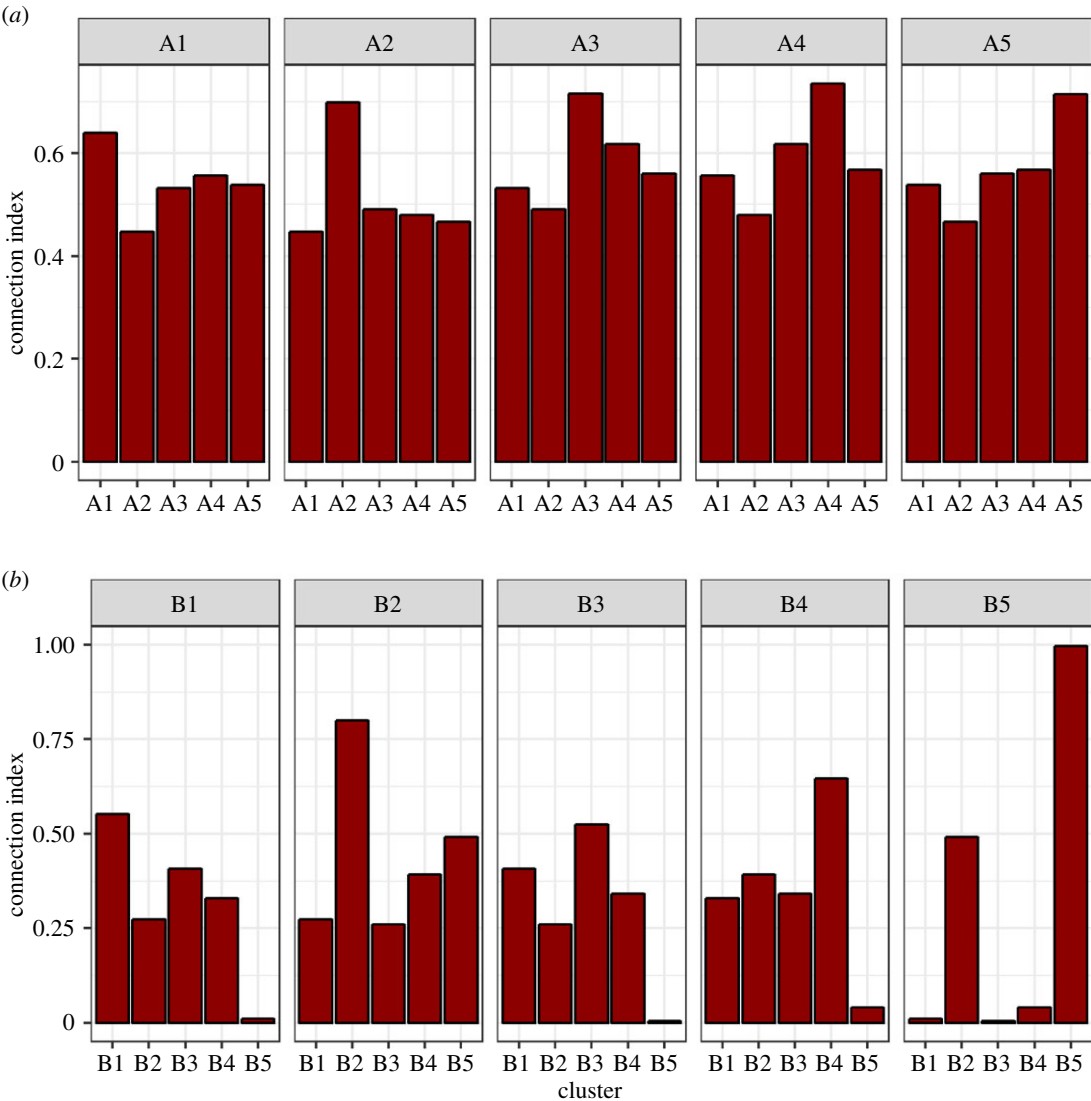

**Figure 3.** Indices of bibliographic connection between papers in the different clusters in (*a*) the period up to and including 2010 and (*b*) the period since 2010. An index of 1 means that all papers are connected by a shared cited reference, and 0 means there are no shared citations between any of the papers in the two clusters. (Online version in colour.)

## (b) Structure of the recent literature (post-2010)

The recent literature had a more linear structure, with relatively few direct bibliographic links between one end and the other (figure 2*b*). Again, there were five clusters. These separated initially on taxonomic lines (see electronic supplementary material, §7 and table S2): birds and non-primate mammals (B1; red); fish, plants and invertebrate animals (B3; blue); and humans (B2, B4 and B5). Within the human clusters, B2 (mid-green) represented developmental and personality psychology, focusing on individual variation and the impacts of childhood psychosocial experience. Cluster B4 (dark green) represented the research of biological and evolutionary anthropologists, often including data from small-scale societies, and in some cases, non-human primates. Finally, a small cluster, B5 (light green), represented a more specific subset of human personality psychology, concerned in particular with the 'dark triad' traits (Machiavellianism, narcissism and psychopathy). The three human-focused clusters consisted of a total of 225 papers (45% of the total). This compares with 115 (23%) of papers belonging to the sole human cluster A2 in the earlier time period.

The topics studied did not obviously separate clusters B1, B3 and B4. B2, with its focus on psychological variation and childhood psychosocial experience, and B5, with its more specific focus on the dark triad personality traits, were distinct from the other clusters in topic as well as taxon. Formal evolutionary models were found in just 3 of the 50 papers sampled from the post-2010 literature (all in cluster B3). Unlike the earlier time period, there was no item common to 10 most cited reference list of all the clusters (electronic supplementary material, table S2). The classic formal models [27] continued to be prominent in the top-cited references of clusters B1, B3 and B4. The top-cited lists from clusters B2 and B5 contained no references to non-human work or to any formal evolutionary models. The ideas of the 'fast–slow continuum' or 'fast and slow strategies' were much more frequent in the later than the earlier time period, being found in 24 of the 50 sampled papers. This was due to the universal (10/10) allusion to these ideas in the sample papers from clusters B2 and B5. However, mention of the ideas remained rare outside of these two clusters.

## (c) Connections within and between clusters

The connection index in the earlier time period was 0.56 overall (i.e. 56% of possible bibliographic links actually existed).

Links were well distributed across clusters (figure 3a): indeed, the papers from all clusters were only modestly more linked to other papers within their cluster than to papers outside their cluster. Reflecting this, the Gini coefficients for the inequality of connections across clusters were low (A1, 0.08; A2, 0.10; A3, 0.09; A4, 0.10; A5, 0.09), and the within/between connection ratios were only slightly greater than 1 (A1, 1.23; A2, 1.48; A3, 1.30; A4, 1.32; A5, 1.34). In the later time period, the connection index overall was 0.35. The distribution of links across clusters was much more uneven (figure 3b). Correspondingly, the Gini coefficients of connection were higher than the earlier time period, especially for cluster B5 (B1, 0.39; B2, 0.29; B3, 0.39; B4, 0.36; B5 0.80), as were the within/between link ratios (B1, 2.16; B2, 2.25; B3, 2.07; B4, 2.34; B5 7.27). Cluster B5 was only connected to the rest of the literature via B2; its connection indices to all other clusters were close to zero. Cluster B2 had rather low connection indices (less than 0.30) to the two non-human clusters B1 and B3. This means that there is relatively little overlap between the citation lists of the papers in cluster B2 and B5 and the citation lists of papers on non-human organisms.

## 4. Discussion

Using an approach based on bibliographic coupling, we documented the changing structure of the 'life-history theory' literature over time. Our cluster analysis suggests that the field has always been divided to some extent along taxonomic lines, with work on birds, non-human mammals, humans and other organisms tending to show some separation in the sources they cite. Between the early period and the more recent literature, the proportion of papers on humans greatly increased, and the mutual separation of the different styles of human research also became more marked. The three human clusters—evolutionary anthropology, developmental/personality psychology and dark triad—appear on the map at successively greater distances from the clusters of non-human papers. The internal structure of the human literature is approximately but not exactly captured by Black et al.'s [40] distinction between 'bio-demographic' studies (i.e. focused on growth and reproduction) and 'psychological' studies (i.e. focused on behavioural or cognitive traits such as prosociality, personality and religiosity, that are argued to be related to life-history strategy): though all 'psychological' approaches are in clusters B2 and B5, some 'bio-demographic' outcomes such as age at menarche are also studied in cluster B2 (e.g. [41]).

Though we detected five clusters in both time periods, there were several lines of evidence that the literature has become more fragmented. Up to 2010, the probability of two papers sharing a commonly cited reference was only modestly higher (around 30%) if they belonged to the same cluster than if they belonged to different ones. Reflecting this, the Gini coefficients of the inter-cluster connection index were low, indicating that every part of the literature was nearly equally connected to every other part through shared citations. There was also overlap in the most commonly cited references of all clusters. In the post-2010 period, the overall probability of any two papers sharing a cited reference was lower, and links were much more concentrated within clusters (the probability of a shared citation

being at least 100% greater for two papers of the same rather than different clusters). The lower overall probability of sharing a common citation was expected: as the literature has become so much larger, and citation lists are finite, the expected amount of overlap should decrease. However, the increasing dominance of within-cluster citation over between-cluster citation is not an inevitable concomitant of the growth in the size of the literature. Instead, this suggests the formation of separate 'demes' of research all referring to their content as life-history theory. This conclusion is backed up an ancillary analysis of the effects of gradually increasing the sensitivity of the cluster detection algorithm (electronic supplementary material, §5). In the earlier time period, the resolution can be up to 0.26 before any discrete clusters are detected, whereas in the more recent time period, a cluster division is detected with a resolution of 0.15. The initial division in both cases is between human and non-human research.

The shape of the map of the literature up to and including 2010 was basically radial. This means that every cluster has a zone of proximity to every other cluster, at the centre of the map. To document this phenomenon another way, the list of 10 top-cited references in every A cluster contained common theoretical resources, notably [27,28]. By contrast, the post-2010 map was rather linear. This means that the papers in the clusters at one end are only connected to those at the other via a series of intermediate links: dark triad research is linked to research in developmental and personality psychology, which is linked to research on human evolution, which is linked to non-human research. Reflecting this, there were no sources found in the 10 top-cited references of clusters B2 and B5 that were found in the top-cited list of any other cluster.

The existence of discrete clusters is not necessarily a bad thing for a research programme. It is quite understandable that researchers are particularly likely to cite prior results (or methodologies) using their study species or a closely related one. This facilitates local adaptation of methods and ideas to a particular application. Some degree of clustering may be advantageous in terms of the generation of conceptual diversity [13–15]. Moreover, in some of the post-2010 work, particularly cluster B5, life-history theory is not the central theme, but an ancillary resource alluded to in framing the question, sometimes mentioned only in keywords. However, very deep clustering does pose questions of nomenclature: how helpful is it that the different clusters all use the same term, life-history theory? If the probability of shared citation across clusters becomes sufficiently low, conceptual evolution becomes independent: at this point, the term 'life-history theory' may no longer have a unique referent. This can lead to so-called jingle fallacies: arguments based on the erroneous assumption that wherever the same name is being used, the same principles are being invoked. Claims about the successes, flaws, predictions or findings of life-history theory then become problematic: one set of ideas can end up receiving credit or responsibility for the outputs of a different one.

The striking division within our post-2010 sample papers is in allusion to the concept of a 'fast–slow' continuum, or 'fast' and 'slow' life-history strategies. Appeal to this concept is universal in papers from clusters B2 and B5. Indeed, within these clusters, the fast–slow continuum is presented as the central defining idea that individuates life-history theory as

opposed to other approaches. In fact, though, the fast–slow continuum construct is rarely mentioned elsewhere in the literature (4 of 30 of the sampled papers from the other B clusters; [42], a paper on birds, is a rarity in framing life-history theory in similar terms to the papers from clusters B2 and B5). Moreover, the fast–slow continuum concept, at least under that name, was rare in the data up to and including 2010. We suggest that the strong focus on the 'fast–slow' construct represents a shift from seeing 'life-history theory' as a kind of methodology (in ecology and evolution), to seeing it as a search for a characteristic empirical pattern (in psychology and social science). This would be an accommodation in the understanding of 'life-history theory' to what other theories typically look like in those disciplines. Notably, the origins of the 'fast–slow' terminology are inductive, arising from empirical research on cross-species patterns of covariation in multiple life-history traits [43,44]. Thus, the fast–slow concept thus arose not from life-history *theory* (in the sense of formal modelling), but as an inductive generalization from comparative *data*. Modellers have risen to the task of linking the inductive fast–slow comparative findings to life-history theory, notably the work of Charnov [45]. However, this theory deals only with the genetic evolution of covariation in biometric and demographic variables such as age at reproduction and size at maturity. Clusters B2 and B5 extend the fast–slow continuum idea well beyond this: from between-species variation to within-species variation; from biometric to behavioural and personality variables; and (in B2 especially) from genetic evolution to individual plasticity based on developmental experience. The validity of those extensions is beyond our scope here, but their theoretical justification is not trivial.

Biologists working on non-human animals have pursued ideas similar to those found in clusters B2 and B5. They tend to do so, however, under the label 'pace of life' or 'pace-of-life syndrome' (for reviews, see [46,47]). Pace-of-life research is typically presented as distinct from life-history theory: a Web of Science search on 'pace-of-life' produces over 400 records, 95% of which are not found by searching for 'life-history theory' (see electronic supplementary material, §1). This separation may be because there is as yet little formal modelling of the evolutionary basis of the key pace-of-life ideas [48]. Pace-of-life research could become more strongly linked to life-history theory in the future if the relevant evolutionary models are developed, for example, via multivariate generalizations of the reaction norm approach [49]. The central elements of the pace-of-life paradigm are: that a fast–slow continuum might exist between individuals as well as between species or populations; that this single continuum might organize variation in suites of different traits, including behavioural, personality and physiological traits; and that the determinants and consequences of being at different positions along such a continuum should be studied [50]. Réale *et al.* [50] point out that the original source of the pace-of-life idea is the 1970s idea of r- versus K-strategists. Black *et al.* [40] make exactly the same point with respect to the human fast–slow psychological paradigm; indeed, early names for that paradigm were 'r–K' or 'differential K' theory [51,52]. It is somewhat ironic that modern biological life-history theory was largely founded on the *rejection* of a simple 'r–K' continuum and its corresponding causal explanations [27,53], and yet in human research, such a pattern is sometimes taken as definitive of 'life-history theory'. We

would argue that the human research in clusters B2 and B5 of our dataset is conceptually closer to the non-human pace-of-life research than it is to 'life-history theory' as that term is typically used in ecology and evolution. However, at present, clusters B2 and B5 and the pace-of-life paradigm are not making much reference to one another. The three most cited pace-of-life papers [50,54,55] are cited, respectively, 5, 0 and 3 times by the papers of cluster B2, and 0, 0 and 0 times by the papers of cluster B5. Conversely, we have found only a single paper framing the study of variation in humans explicitly within the pace-of-life terminology [56].

Our sampling of papers from each cluster suggested there has been a recent decline in the proportion of the literature that consists of formal models. Up to and including 2010, 16 of 50 papers we sampled presented formal models, a proportion very close to the 30% that Stearns estimated back in 1980 [32]. In the post-2010 sample, we found just three of 50 papers (6%) presenting formal models. This decline was not just due to the absence of formal modelling in the human-focused clusters: 3 of 20 papers from the non-human clusters suggest a reduction in the share of formal modelling there too. It is not clear what might explain this decline. Researchers may believe that the fundamental theory has already been thoroughly mapped out, and what remains is to test its assumptions and predictions, or elucidate the proximate mechanisms underlying trade-offs. Alternatively, the growth in annual productivity of the research programme may be due to more empiricists being attracted to draw on the theoretical ideas, with the absolute number of theoreticians remaining constant, producing a proportionate decline in theory papers. These remain speculations on our part.

It is possible that the decline in formal modelling, or in direct reliance on the results of such modelling, has exacerbated fragmentation. Theories stated in natural language can readily mutate through qualitative interpretation and memorial bias, so that their transmission resembles a game of telephone [57]. For example, citations in papers are quite often (perhaps 15% of instances) used in support of statements that are substantively different from, absent from, or even contradictory to, the statements in the paper cited [58,59]. When a theory is formalized in mathematical models, however, it may be transmitted with greater fidelity [57]. Consider for example, Newtonian mechanics, or the modern synthesis version of evolutionary theory. New discoveries may prove challenging to accommodate within these bodies of theory, but scientists at least all agree what the bodies of theory consist of. We would predict that research programmes based closely on formal theory will tend to remain more stable and conceptually unified and that a shift to more verbally based theory will also lead to more rapid conceptual change and greater fragmentation. These predictions could be tested in a future study.

We should acknowledge the limitations of our study. First, as mentioned at the outset, ours is a study of the use of the label 'life-history theory', and not a comprehensive review of all of the literature on life-history evolution. Much canonical work on the evolution of life-history traits does not use our exact search terms and hence is not captured. In the electronic supplementary material, §1, we do undertake a broader search strategy. This returns almost four times as many documents, including much more research on plants, invertebrate animals and fish than

'life-history theory'. The detailed cluster structure of this broader literature is thus somewhat different. However, several of our conclusions for 'life-history theory' appear to hold for this broader literature too. Specifically, map structure is radial in the earlier time period and more linear in the recent one; and the human research in the recent time period is concentrated at one end of the continuum, relatively separate from the rest of the literature. Second, in creating bibliometric maps in VOS viewer, there are multiple user-settable parameters. We have generally followed precedent in choosing values for these, but slightly different choices affect the results. In particular, the cluster divisions between clusters B2, B4 and B5 are sensitive to parameter settings; small changes result in one or more of these divisions not being detected (see electronic supplementary material, §§2, 3 and 5 for examples). This obviously affects subsequent calculations like indices of connection. While cluster detection is sensitive to parameter values, the overall difference in shape between the early and later period maps appears to be very robust. Finally, for our investigation of the content of research areas, we have only sampled 10 papers from each cluster. While this gives sufficient evidence for describing broad differences between clusters, sampling only 10 papers will underestimate the intellectual diversity within each cluster. Each of our clusters could itself be the subject of a detailed qualitative review.

In conclusion, we have documented the evolution over time of the literature appealing to the concept of life-history theory. This literature has grown rapidly and increasingly incorporated more psychological and social-science research. As it has grown, its internal differentiation has become more marked. It may be incipiently speciating into a part concerned with a fast–slow continuum of mainly psychological traits in humans, a part we argue is conceptually allied with the pace-of-life hypothesis; and a body of work that shares the original concerns of life-history evolution, drawing more heavily on formal modelling.

Data accessibility. A comprehensive data archive is available at http://doi.org/10.5281/zenodo.2530683. This includes the raw Web of Science records, the lists of papers in each cluster, the VOS viewer files and the R code required to reproduce the analyses. Readers can also explore interactive high-resolution versions of figure 2a,b by typing the URLs tiny.cc/LHTearlymap and tiny.cc/LHTrecent-map into a Web browser (requires Java, may involve downloading and opening a file).

Competing interests. We declare we have no competing interests.

Funding. This project has received funding from the European Research Council (ERC) under the European Union's Horizon 2020 research and innovation programme (grant agreement no. AdG 666669, COM-STAR, to D.N.); and by grants from the Netherlands Organization for Scientific Research (016.155.195), the James S. McDonnell Foundation (220020502), the Jacobs Foundation (2017 1261 02) and the Robert Wood Johnson Foundation (73657) to W.E.F.

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
