## [Reviewer comments · Proceedings of the Royal Society B: Biological Sciences]

Review History

RSPB-2019-0040.R0 (Original submission)

Review form: Reviewer 1

Recommendation

Accept with minor revision (please list in comments)

Do you want your name to be published alongside your report?

Yes

If yes, please enter your name here as it should appear.

Stephen C. Stearns

Scientific importance: Is the manuscript an original and important contribution to its field?

Excellent

General interest: Is the paper of sufficient general interest?

Excellent

Quality of the paper: Is the overall quality of the paper suitable?

Good

Additional comments to the Editor:

This is a detailed and measured assessment of the spread of concepts through science. It should become one of the yardsticks by which the power of bibliometric analysis gets measured. The authors need to check the robustness of their main results to changes in their search term.

Is the length of the paper justified?

Yes

Do you think some of the material would be more appropriate as an electronic appendix?

No

Should the paper be seen by a specialist statistical reviewer?

No

Do you have any concerns about statistical analyses in this paper? If so, please specify them explicitly in your report.

No

Is it accessible?

Yes

Is it clear?

Yes

Is it adequate?

Yes

Do you have any ethical concerns with this paper? Please include details of any concerns in the comments to Editors section below.

No

Additional comments to the Editor:

I do have concerns with the ethics of some of the literature that they analyze, but I do not have any concerns with the ethics of this paper.

Comments to the Author

There was a lot of life history theory published before Charnov (1979) that this search is not picking up. The most significant conceptual seed was Williams (1966a,b), both the book and the paper. A key publication presenting formal theory was Gadgil & Bossert (1970). Did your search reveal references to Brian Charlesworth's papers and book? A citation analysis that misses those sources is suspect and reinforces the need to strengthen the analysis by examining how robust the results are to the formulation of the search term. Would the results change significantly if the search had been performed using "life history evolution" or "life history strategy" rather than "life history theory"?

Line 124: cut "were" - a mistake left from editing.

Line 138: "taxa from which" not "taxa from whom", "data are" not "data is"

Line 196: One recalls the adage that a work or a concept can become so well known that it is

accepted as part of the general background of a field and no longer needs to be cited because “everybody knows that”. Perhaps you are measuring roughly when this happened to “life history theory”: about 2010.

Line 215: Cluster B5 post-2010 appears to be an echo chamber in which fake news reflects off relatively impermeable walls. See my comment about the personal impact of your analysis at the end of this review.

Line 236: The deep and important distinction between the analysis of causes and the description of patterns is nowhere more evident than in the superficiality suggested by the description of what is going on in post-2010 clusters B2 and B5. One would at least hope that instead of simply invoking a biological substrate for social and psychological patterns that these folks would explicitly consider the possibility that there might be purely cultural processes that could generate all or part of the patterns described. That is, of course, a problem with their papers, not with your analysis, which does a pretty good job of revealing what is going on. You are perfectly correct to raise the issue, what does the word theory mean in that literature? Evidently it is more of an excuse to make one comfortable than an explanation of an important piece of nature. Here let me recall John Maynard Smith’s dichotomization of scientists. There are two kinds, he said. Those who want to be right, and those who want to know the truth. If you want to know the truth, you have to be prepared to admit when you are wrong.

Line 280: The controversy about the meaning of r&K-selection in the late 1970’s has strong parallels to the contrast you draw between descriptions of slow-fast patterns and theoretical analyses of the evolutionary causes that produce those patterns. Here again the use of particular search terms misses the origin of concepts, which evidently go through a period of incubation before being fossilized in a particular form. The slow-fast continuum, earlier referred to as r&K selection by MacArthur and Wilson (1967) and Pianka (1970) was quantified as Principal Component 1 in Stearns, S.C. 1983. The influence of size and phylogeny on patterns of covariation in the life-history traits of mammals. *Oikos* 41: 173-87. The fact that the percentage of total variation that it captured dropped from 68-75% to 29-36% when order and family effects were removed tells us a lot about the portion of variation that theory was explaining, for theory did not include the clade-specific effects that one might call phylogenetic constraint or inertia. The pattern being invoked, in other words, was more complicated and not as strong as those working in areas B2 and B5 appear to believe and had causes that were at least as deeply rooted in the ancient and inherited design of large groups of organisms as in any immediate selection produced by variation in demographic vital rates. George Santayana: “Those who cannot remember the past are condemned to repeat it.” (Some think we are condemned to repeat it whether we remember it or not, but I would cite the existence of the EU and UN, however imperfect they may be, as evidence that counts against that claim.)

Lines 289-290: spot on.

Lines 298-319: See comments on Line 280. In addition, one could point out that one concept that connects life history theory to pace of life discussions is that of adaptive reaction norms. One can conceptually extend the very well analyzed idea of reaction norms for age and size at maturity, now commonplace in the fisheries literature, to reaction surfaces for syndromes of life history traits, which is what pace of life describes.

Lines 333-335: You make very good points about the role that a core body of theory plays in stabilizing a field. It also plays an essential role in checking the logic of claims and reducing the possibility of error introduced by the kind of story-telling that aims at confirming pet hypotheses rather than demanding that claims about reality stand up to strong confrontation with alternative hypotheses. Consider adding a comment to that effect.

I found myself quite emotionally engaged with this paper because it recalls issues that both shaped and threatened my career in science. When I abandoned r&K selection as an organizing principle after having used it in my 1976 review paper, which was based on my 1973 PhD proposal, and pointed out, along with Brian Charlesworth (who was ahead of me on this), that the analysis of the evolutionary consequences of age-specific variation in birth and death rates got us a lot closer to causes than did coarse-grained analogies to the Lotka-Volterra model of density dependent population regulation, I made an enemy of E.O. Wilson for life. Fortunately, he was not the only influential figure in the field, and George Williams became a supporter and a friend.

I would also like to register that I am deeply offended at the way that some evolutionary psychologists have used references to life history theory to support claims that pace of life and slow-fast analogies explain variation in tendencies towards criminality. Intellectual dishonesty often contributes to moral failure. See, for example, Peter Rosenthal's paper, "The criminality of racial harassment," (1989-1990 Can. Hum. Rts. Y.B. 113 (1989-1990) which comments on Phillippe Rushton's use of r&K selection to support racist propaganda.

Review form: Reviewer 2

Recommendation

Accept with minor revision (please list in comments)

Do you want your name to be published alongside your report?

No

Scientific importance: Is the manuscript an original and important contribution to its field?

Good

General interest: Is the paper of sufficient general interest?

Marginal

Quality of the paper: Is the overall quality of the paper suitable?

Good

Is the length of the paper justified?

Yes

Do you think some of the material would be more appropriate as an electronic appendix?

No

Should the paper be seen by a specialist statistical reviewer?

No

Do you have any concerns about statistical analyses in this paper? If so, please specify them explicitly in your report.

No

Is it accessible?

Yes

Is it clear?

Yes

Is it adequate?

Yes

Do you have any ethical concerns with this paper? Please include details of any concerns in the comments to Editors section below.

No

Comments to the Author

This manuscript provides valuable insight into the intellectual structure of life history theory, which appears to be timely given it's increased use in the social sciences. I particularly liked how the researchers split the network into two time periods to examine structural changes through time. Overall, I think that I would recommend this manuscript for publication in Proceedings B, but first I would like to see some revisions and clarifications to the methods.

The introduction provides sufficient background on both life history theory and bibliometric methods, but the aims and hypotheses are scattered in a way that is hard to follow. The first aim appears in lines 51-54, followed by two paragraph on bibliometrics, followed by another aim on lines 85-86, and then a hypothesis at the end of the last paragraph. This should be restructured so that the aims and hypotheses appear in the same location. Additionally, the logic of the last hypothesis in the last paragraph of the intro seems weak compared to the justification in the discussion. Try to make the distinction between the hypothesized differences between biology and the social sciences more explicit.

The methods include sufficient detail to replicate the study, and the most important research decisions (i.e. bibliographic coupling over co-citation) are well-justified. I'm also happy to see that the researchers included the results of different cluster resolutions in the supporting information. I think clarification is needed in a couple of places, though. For example, why was the default VOSviewer cut-off of 500 documents (see line 124) chosen rather than the largest connected network? Depending on network size in each time period, this decision could skew the measured levels of connectivity. Using only 10 sample papers from each cluster also seems quite low to me. For larger clusters, less of the intellectual diversity will be characterized. This limitation should at least be acknowledged in the discussion. Lines 144-148 are a bit unclear. Were the proportion of possible links calculated for each pair of clusters? Or just within clusters? The Gini coefficient appears to be an appropriate measure for inequality in connections.

At the beginning of the results, the authors say that after "an initial phase of exponential growth, the number of new papers has grown linearly since around 2005". It seems in the figure that an exponential function would fit the data well, so it might be worth replacing the loess fit with that and changing the wording in the results.

Figure 2 uses the same colors for both maps, which is briefly mentioned in the caption but should be expanded. I think that most readers will mistakenly assume that the clusters are analogous between time periods, so the researchers should take extra care to clarify this.

In the discussion (see lines 232-233) the researchers mention that "the three human clusters... lie at progressively greater bibliometric distance from life history theory as it is practiced in ecology

and evolutionary biology". Were the distances between clusters actually calculated, or is this just a qualitative conclusion?

There are some minor cases of grammar mistakes and run-on sentences that should be addressed. For example...

- Line 51: "become different kinds of thing?"
- Line 104: "of the search term in title, abstract..."
- Line 124: "We created maps were based..."
- Lines 192-194: Sentence could be broken up
- Lines 340-342: Sentence could be broken up

Decision letter (RSPB-2019-0040.R0)

04-Feb-2019

Dear Professor Nettle:

Your manuscript has now been peer reviewed and the reviews have been assessed by an Associate Editor. The reviewers' comments (not including confidential comments to the Editor) and the comments from the Associate Editor are included at the end of this email for your reference. As you will see, the reviewers and the Editors have raised some concerns with your manuscript and we would like to invite you to revise your manuscript to address them.

Research ethics:

Use of animals and field studies:

Please submit a copy of your revised paper within three weeks. If we do not hear from you within this time your manuscript will be rejected. If you are unable to meet this deadline please let us know as soon as possible, as we may be able to grant a short extension.

Best wishes,
 Proceedings B
 mailto: proceedingsb@royalsociety.org

Associate Editor
 Board Member: 1
 Comments to Author:

I have now received two careful and constructive reviews for your manuscript "The evolution of life history theory: Bibliometric analysis of an interdisciplinary research area". Both the reviewers and myself enjoyed reading this manuscript, and found it could be an interesting publication for Proc B. Although the reviews were overall positive there were a few important issues raised by both reviewers that could be addressed and likely improve the paper. Namely, Reviewer 1 suggests some reanalysis of the data using alternate search terms, which I wholeheartedly agree with; and Reviewer 2 suggests a reorganization of the Introduction in a manner that better clarifies the logic and aims of the manuscript to improve readability. I hope that you find the comments of the reviewers helpful in revising your manuscript.

Reviewer(s)' Comments to Author:

Referee: 1

Comments to the Author(s)

There was a lot of life history theory published before Charnov (1979) that this search is not picking up. The most significant conceptual seed was Williams (1966a,b), both the book and the paper. A key publication presenting formal theory was Gadgil & Bossert (1970). Did your search reveal references to Brian Charlesworth's papers and book? A citation analysis that misses those sources is suspect and reinforces the need to strengthen the analysis by examining how robust the results are to the formulation of the search term. Would the results change significantly if the search had been performed using "life history evolution" or "life history strategy" rather than "life history theory"?

Line 124: cut "were" - a mistake left from editing.

Line 138: "taxa from which" not "taxa from whom", "data are" not "data is"

Line 196: One recalls the adage that a work or a concept can become so well known that it is accepted as part of the general background of a field and no longer needs to be cited because "everybody knows that". Perhaps you are measuring roughly when this happened to "life history theory": about 2010.

Line 215: Cluster B5 post-2010 appears to be an echo chamber in which fake news reflects off relatively impermeable walls. See my comment about the personal impact of your analysis at the end of this review.

Line 236: The deep and important distinction between the analysis of causes and the description of patterns is nowhere more evident than in the superficiality suggested by the description of what is going on in post-2010 clusters B2 and B5. One would at least hope that instead of simply invoking a biological substrate for social and psychological patterns that these folks would

explicitly consider the possibility that there might be purely cultural processes that could generate all or part of the patterns described. That is, of course, a problem with their papers, not with your analysis, which does a pretty good job of revealing what is going on. You are perfectly correct to raise the issue, what does the word theory mean in that literature? Evidently it is more of an excuse to make one comfortable than an explanation of an important piece of nature. Here let me recall John Maynard Smith's dichotomization of scientists. There are two kinds, he said. Those who want to be right, and those who want to know the truth. If you want to know the truth, you have to be prepared to admit when you are wrong.

Line 280: The controversy about the meaning of r&K-selection in the late 1970's has strong parallels to the contrast you draw between descriptions of slow-fast patterns and theoretical analyses of the evolutionary causes that produce those patterns. Here again the use of particular search terms misses the origin of concepts, which evidently go through a period of incubation before being fossilized in a particular form. The slow-fast continuum, earlier referred to as r&K selection by MacArthur and Wilson (1967) and Pianka (1970) was quantified as Principal Component 1 in Stearns, S.C. 1983. The influence of size and phylogeny on patterns of covariation in the life-history traits of mammals. *Oikos* 41: 173-87. The fact that the percentage of total variation that it captured dropped from 68-75% to 29-36% when order and family effects were removed tells us a lot about the portion of variation that theory was explaining, for theory did not include the clade-specific effects that one might call phylogenetic constraint or inertia. The pattern being invoked, in other words, was more complicated and not as strong as those working in areas B2 and B5 appear to believe and had causes that were at least as deeply rooted in the ancient and inherited design of large groups of organisms as in any immediate selection produced by variation in demographic vital rates. George Santayana: "Those who cannot remember the past are condemned to repeat it." (Some think we are condemned to repeat it whether we remember it or not, but I would cite the existence of the EU and UN, however imperfect they may be, as evidence that counts against that claim.)

Lines 289-290: spot on.

Lines 298-319: See comments on Line 280. In addition, one could point out that one concept that connects life history theory to pace of life discussions is that of adaptive reaction norms. One can conceptually extend the very well analyzed idea of reaction norms for age and size at maturity, now commonplace in the fisheries literature, to reaction surfaces for syndromes of life history traits, which is what pace of life describes.

Lines 333-335: You make very good points about the role that a core body of theory plays in stabilizing a field. It also plays an essential role in checking the logic of claims and reducing the possibility of error introduced by the kind of story-telling that aims at confirming pet hypotheses rather than demanding that claims about reality stand up to strong confrontation with alternative hypotheses. Consider adding a comment to that effect.

I found myself quite emotionally engaged with this paper because it recalls issues that both shaped and threatened my career in science. When I abandoned r&K selection as an organizing principle after having used it in my 1976 review paper, which was based on my 1973 PhD proposal, and pointed out, along with Brian Charlesworth (who was ahead of me on this), that the analysis of the evolutionary consequences of age-specific variation in birth and death rates got us a lot closer to causes than did coarse-grained analogies to the Lotka-Volterra model of density dependent population regulation, I made an enemy of E.O. Wilson for life. Fortunately,

he was not the only influential figure in the field, and George Williams became a supporter and a friend.

I would also like to register that I am deeply offended at the way that some evolutionary psychologists have used references to life history theory to support claims that pace of life and slow-fast analogies explain variation in tendencies towards criminality. Intellectual dishonesty often contributes to moral failure. See, for example, Peter Rosenthal's paper, "The criminality of racial harassment," (1989-1990 Can. Hum. Rts. Y.B. 113 (1989-1990) which comments on Phillippe Rushton's use of r&K selection to support racist propaganda.

Referee: 2

Comments to the Author(s)

This manuscript provides valuable insight into the intellectual structure of life history theory, which appears to be timely given it's increased use in the social sciences. I particularly liked how the researchers split the network into two time periods to examine structural changes through time. Overall, I think that I would recommend this manuscript for publication in Proceedings B, but first I would like to see some revisions and clarifications to the methods.

The introduction provides sufficient background on both life history theory and bibliometric methods, but the aims and hypotheses are scattered in a way that is hard to follow. The first aim appears in lines 51-54, followed by two paragraph on bibliometrics, followed by another aim on lines 85-86, and then a hypothesis at the end of the last paragraph. This should be restructured so that the aims and hypotheses appear in the same location. Additionally, the logic of the last hypothesis in the last paragraph of the intro seems weak compared to the justification in the discussion. Try to make the distinction between the hypothesized differences between biology and the social sciences more explicit.

The methods include sufficient detail to replicate the study, and the most important research decisions (i.e. bibliographic coupling over co-citation) are well-justified. I'm also happy to see that the researchers included the results of different cluster resolutions in the supporting information. I think clarification is needed in a couple of places, though. For example, why was the default VOSviewer cut-off of 500 documents (see line 124) chosen rather than the largest connected network? Depending on network size in each time period, this decision could skew the measured levels of connectivity. Using only 10 sample papers from each cluster also seems quite low to me. For larger clusters, less of the intellectual diversity will be characterized. This limitation should at least be acknowledged in the discussion. Lines 144-148 are a bit unclear. Were the proportion of possible links calculated for each pair of clusters? Or just within clusters? The Gini coefficient appears to be an appropriate measure for inequality in connections.

At the beginning of the results, the authors say that after "an initial phase of exponential growth, the number of new papers has grown linearly since around 2005". It seems in the figure that an exponential function would fit the data well, so it might be worth replacing the loess fit with that and changing the wording in the results.

Figure 2 uses the same colors for both maps, which is briefly mentioned in the caption but should be expanded. I think that most readers will mistakenly assume that the clusters are analogous between time periods, so the researchers should take extra care to clarify this.

In the discussion (see lines 232-233) the researchers mention that "the three human clusters... lie at progressively greater bibliometric distance from life history theory as it is practiced in ecology

and evolutionary biology". Were the distances between clusters actually calculated, or is this just a qualitative conclusion?

There are some minor cases of grammar mistakes and run-on sentences that should be addressed. For example...

- Line 51: "become different kinds of thing?"
- Line 104: "of the search term in title, abstract..."
- Line 124: "We created maps were based..."
- Lines 192-194: Sentence could be broken up
- Lines 340-342: Sentence could be broken up

Author's Response to Decision Letter for (RSPB-20190040.R0)

See Appendix A.

Decision letter (RSPB-2019-0040.R1)

07-Mar-2019

Dear Professor Nettle

I am pleased to inform you that your manuscript RSPB-2019-0040.R1 entitled "The evolution of life history theory: Bibliometric analysis of an interdisciplinary research area" has been accepted for publication in Proceedings B.

The referee(s) have recommended publication, but also suggest some minor revisions to your manuscript. Therefore, I invite you to respond to the referee(s)' comments and revise your manuscript. Because the schedule for publication is very tight, it is a condition of publication that you submit the revised version of your manuscript within 7 days. If you do not think you will be able to meet this date please let us know. Only very minor changes are needed.

Sincerely,
Professor John R Hutchinson
Editor
Proceedings B
<mailto:proceedingsb@royalsociety.org>

Author's Response to Decision Letter for (RSPB-20190040.R1)

See Appendix B.

Decision letter (RSPB-2019-0040.R2)

08-Mar-2019

Dear Professor Nettle

I am pleased to inform you that your manuscript entitled "The evolution of life history theory: Bibliometric analysis of an interdisciplinary research area" has been accepted for publication in Proceedings B.

Open Access

Paper charges

Sincerely,

Proceedings B
mailto:proceedingsb@royalsociety.org

Appendix A

Response to referees

Thank you for helpful comments on this paper, which we have enjoyed responding to. The main issues you highlighted were, first, analysis of alternate search terms. We have now done this, and report it in the revised section 1 of the supporting information, also drawing attention to it in the discussion, and explaining at the outset why we used the search term 'life history theory' for the main part of the paper (see first response to referee 1). Second, reorganization of the introduction to make the logic and aims clearer. We have rewritten the introduction to consolidate the aims at the end, as suggested (see first response to referee 2).

In addition, we have made a number of more minor changes, as detailed below. For the comments of referee 1, there is a lot more than could be said on the history of life history theory and the validity of some of its extensions to humans. However, in view of the nature of the paper and *Proceedings* word length, we have restricted ourselves to a few brief extra points in the discussion.

Referee: 1

There was a lot of life history theory published before Charnov (1979) that this search is not picking up.

The most significant conceptual seed was Williams (1966a,b), both the book and the paper. A key

publication presenting formal theory was Gadgil & Bossert (1970). Did your search reveal references to

Brian Charlesworth's papers and book? A citation analysis that misses those sources is suspect and

reinforces the need to strengthen the analysis by examining how robust the results are to the

formulation of the search term. Would the results change significantly if the search had been

performed using "life history evolution" or "life history strategy" rather than "life history theory"?

Thanks for these suggestions. To be clear, the object of our study was the particular label 'life history theory', since this has become a very recognisable 'badge' in both biology and psychology/social science. We are principally interested in the evolution of the use of this label, rather than a full bibliometric review of the field of life history evolution. It is clear that searching for the 'life history theory' label does not detect the entire literature on life history evolution, as the referee rightly states. In the early decades, people did not use this exact label. However, we have now also done a larger search: "'life history theory' OR 'life history strategies' OR 'life history evolution'." We report the results of the broader search in the Supporting Information, section 1.

The broader search returns four times as many records as our 'life history theory' search. It captures more of the early work, though still not all of it, which suggests that digitisation

on Web of Science is patchy in the 1970s. It captures a great deal more work on plants and fishes, and the cluster structure and shape of the map are somewhat different from the analyses reported in the main paper (see Supporting Information, section 1). However, the central point remains in this broader analysis: there has been a large growth in human psychological/social work that alludes to life history theory/strategies/evolution; and this work is in its own cluster(s), rather isolated from the non-human literatures.

We have taken the following actions to make our aims clearer and our analysis more robust:

1. Rewritten the introduction to clarify that we aimed to study the evolution of the particular label 'life history theory', even though this exact label was not usually used in the seminal early work, and that we are not claiming to have reviewed all or even a representative sample of all literature on life history evolution.
2. Expanded and modified section 1 of the Supporting Information to cover the broader search described above and present the main results of analysing that search.
3. Alluded to this limitation in the penultimate paragraph of the discussion, briefly mentioning the conclusions that the broader search leads to, and how these differ from the narrower one 'life history theory'.

Line 124: cut "were" – a mistake left from editing.

Thanks, corrected.

Line 138: "taxa from which" not "taxa from whom", "data are" not "data is"

Corrected.

Line 196: One recalls the adage that a work or a concept can become so well known that it is accepted

as part of the general background of a field and no longer needs to be cited because "everybody knows

that". Perhaps you are measuring roughly when this happened to "life history theory": about 2010.

Well, if that is so, it only appears to be so amongst people who work on humans, since Stearns' *Evolution of Life Histories*, and earlier work such as George C. Williams, continues to be highly cited in the clusters B1 and B3 where people are working on non-human species. We suspect, rather, that this is the product of people who work on humans relying on secondary descriptions of what life-history theory is. Therefore, the lack of citation to a common core of work is likely to reflect different people getting their secondary descriptions from different sources. We haven't actually made a change to the paper on this point – exactly what is going on is something of a speculation.

Line 215: Cluster B5 post-2010 appears to be an echo chamber in which fake news reflects off relatively

impermeable walls. See my comment about the personal impact of your analysis at the end of this review.

We might well agree with the reviewer here—at any rate, this is a cluster of work that refers to itself in a relatively insular way. But in this paper we are trying to present the data in an objective way, without any value judgements about any of the work captured by the searches. We will leave readers to make their own inferences about the scientific dynamics in the various clusters.

Line 236: The deep and important distinction between the analysis of causes and the description of patterns is nowhere more evident than in the superficiality suggested by the description of what is going on in post-2010 clusters B2 and B5. One would at least hope that instead of simply invoking a biological substrate for social and psychological patterns that these folks would explicitly consider the possibility that there might be purely cultural processes that could generate all or part of the patterns described. That is, of course, a problem with their papers, not with your analysis, which does a pretty good job of revealing what is going on. You are perfectly correct to raise the issue, what does the word theory mean in that literature? Evidently it is more of an excuse to make one comfortable than an explanation of an important piece of nature. Here let me recall John Maynard Smith's dichotomization of scientists. There are two kinds, he said. Those who want to be right, and those who want to know the truth. If you want to know the truth, you have to be prepared to admit when you are wrong.

This is a really interesting comment. As the referee says, it is a critique of the work we are reviewing more than of our review. We have, however, sharpened up our distinction between the description of patterns and the analysis of causes, because we think it is useful in terms of understanding what is going on in the different parts of the literature. In particular, it seems right to say that biological 'life history theory' research is typically concerned with the analysis of causes, whereas the human research is more typically concerned with the description of patterns. We set up the expectation that this may be the case in the revised introduction:

By theory, Stearns [32] meant the practice of formal mathematical modelling of fitness in relation to life history traits, rather than any particular empirical claim that might arise from such models. However, in psychology and most social sciences, the term 'theory' refers to frameworks that are not formalized, and are at least to some extent inductive (based on typical patterns in data) rather than deductive (based on logical inferences from axioms). If 'life history theory' has adapted to the substrate of psychology and the social sciences as it has spread, we should expect decreasing use of formal models, and an increasingly close association of 'life history theory' with a characteristic empirical pattern. Our impression (from, for example [4]) is that this has

indeed happened. In many psychological and social-science works, 'life history theory' is used as a near-synonym for the 'fast-slow continuum' (or 'fast' and 'slow' strategies). This is the descriptive generalization that variation between species or between individuals can be organized onto a principal axis from early maturation and reproduction, small body size, large numbers of offspring and low parental investment at one end; to late maturation and reproduction, large body size, small numbers of offspring and high parental investment at the other. A full review of the sources, varieties of, and evidence for, the 'fast-slow continuum' idea is beyond our scope here. We merely hypothesise that it might play a different role in 'life history theory' in psychology and social science as compared to ecology and evolution.

We then return to this distinction in the discussion, in the passage on the widespread appeal to the 'fast-slow' continuum idea in psychology and the social sciences:

We suggest that the strong focus on the 'fast-slow' construct represents a shift from seeing 'life history theory' as a kind of methodology (in ecology and evolution), to seeing it as search for a characteristic empirical pattern (in psychology and social science). This would be an accommodation in the understanding of 'life history theory' to what other theories typically look like in those disciplines. Notably, the origins of the 'fast-slow' terminology are inductive, arising from empirical research on cross-species patterns of covariation in multiple life history traits [43,44]. Thus, the fast-slow concept thus arose not from life history theory (in the sense of formal modelling), but as an inductive generalization from comparative data.

Line 280: The controversy about the meaning of r&K-selection in the late 1970's has strong parallels to the contrast you draw between descriptions of slow-fast patterns and theoretical analyses of the evolutionary causes that produce those patterns. Here again the use of particular search terms misses the origin of concepts, which evidently go through a period of incubation before being fossilized in a particular form. The slow-fast continuum, earlier referred to as r&K selection by MacArthur and Wilson (1967) and Pianka (1970) was quantified as Principal Component 1 in Stearns, S.C. 1983. The influence of size and phylogeny on patterns of covariation in the life-history traits of mammals. Oikos 41: 173-87. The fact that the percentage of total variation that it captured dropped from 68-75% to 29-36% when order and family effects were removed tells us a lot about the portion of variation that theory was explaining, for theory did not include the clade-specific effects that one might call phylogenetic

constraint or inertia. The pattern being invoked, in other words, was more complicated and not as strong as those working in areas B2 and B5 appear to believe and had causes that were at least as deeply rooted in the ancient and inherited design of large groups of organisms as in any immediate selection produced by variation in demographic vital rates. George Santayana: "Those who cannot remember the past are condemned to repeat it." (Some think we are condemned to repeat it whether we remember it or not, but I would cite the existence of the EU and UN, however imperfect they may be, as evidence that counts against that claim.)

Thanks for this. This is useful background. In fact, reading what psychologists write when they talk about 'fast-slow', it seems that they have 'r/K' strategists out of Pianka (1970) pretty squarely in mind. It is thus ironic that those models fell out of view so long ago in biology, and yet psychologists are still adopting the idea as if it were biological orthodoxy. Length constrains us from going into this in too much detail here. In the discussion of the 'pace of life' paradigm and human life history theory, we have however added the sentence:

(Indeed, it is somewhat ironic that modern biological life history theory was largely founded on the rejection of a simple 'r-K' continuum and its corresponding causal explanations [27,53], and yet in human research, such a pattern is sometimes taken as definitive of 'life history theory').

Lines 289-290: spot on.

Thank you.

Lines 298-319: See comments on Line 280. In addition, one could point out that one concept that connects life history theory to pace of life discussions is that of adaptive reaction norms. One can conceptually extend the very well analyzed idea of reaction norms for age and size at maturity, now commonplace in the fisheries literature, to reaction surfaces for syndromes of life history traits, which is what pace of life describes.

We have added a brief reference to this point, with a citation to Stearns and Koella (1986) on reaction norms in understanding life history variation.

Lines 333-335: You make very good points about the role that a core body of theory plays in stabilizing a field. It also plays an essential role in checking the logic of claims and reducing the possibility of error introduced by the kind of story-telling that aims at confirming pet hypotheses rather than demanding

that claims about reality stand up to strong confrontation with alternative hypotheses. Consider adding a comment to that effect.

This strikes us a potentially complex issue. There are some areas (cultural evolution theory springs to mind) where having a body of core formal theory has led people to look for the phenomena assumed or predicted in the formal models, rather than really describing how cultural phenomena actually work. So the point about confirmation bias appears to us to be orthogonal to the issue of having formal core theory or not.

I found myself quite emotionally engaged with this paper because it recalls issues that both shaped and threatened my career in science. When I abandoned r&K selection as an organizing principle after having used it in my 1976 review paper, which was based on my 1973 PhD proposal, and pointed out, along with Brian Charlesworth (who was ahead of me on this), that the analysis of the evolutionary consequences of age-specific variation in birth and death rates got us a lot closer to causes than did coarse-grained analogies to the Lotka-Volterra model of density dependent population regulation, I made an enemy of E.O. Wilson for life. Fortunately, he was not the only influential figure in the field, and George Williams became a supporter and a friend.....I would also like to register that I am deeply offended at the way that some evolutionary psychologists have used references to life history theory to support claims that pace of life and slow-fast analogies explain variation in tendencies towards criminality. Intellectual dishonesty often contributes to moral failure. See, for example, Peter Rosenthal's paper, "The criminality of racial harassment," (1989-1990 Can. Hum. Rts. Y.B. 113 (1989-1990) which comments on Phillippe Rushton's use of r&K selection to support racist propaganda.

Thanks for these comments, especially on your view of some of the human research. We are now working on a more qualitative paper on 'life history theory' as it is understood in the human sciences. In that paper, we would love to explore the issues a bit more critically. In this paper, we are trying to stay focussed on the objective bibliometrics, and so we have not made any major change in response to the above. We have added to the discussion the following brief point:

Clusters B2 and B5 extend the fast-slow continuum idea....from between-species variation to within-species variation; from biometric to behavioural and personality variables; and (in B2 especially) from genetic evolution to individual plasticity based on developmental experience. The validity of those extensions is beyond our scope here, but their theoretical justification is not trivial.

Referee: 2

The introduction provides sufficient background on both life history theory and bibliometric methods, but the aims and hypotheses are scattered in a way that is hard to follow. The first aim appears in lines 51-54, followed by two paragraph on bibliometrics, followed by

another aim on lines 85-86, and then a hypothesis at the end of the last paragraph. This should be restructured so that the aims and hypotheses appear in the same location.

We have restructured the introduction so that the aims are consolidated at the end, instead of being peppered throughout. To facilitate this, we have moved one paragraph (a relatively detailed one on how bibliometric analysis is done) out of the introduction to the first paragraph of the methods. We hope that the aims and hypotheses are clearer now.

Additionally, the logic of the last hypothesis in the last paragraph of the intro seems weak compared to the justification in the discussion. Try to make the distinction between the hypothesized differences between biology and the social sciences more explicit.

We have tried to flesh out the logic of our hypothesis without pre-empting the findings by expanding our discussion of what ‘theories’ typically look like in psychology and social science (see penultimate paragraph of revised introduction). A typical psychological theory is based on identifying a *characteristic pattern* rather than a priori prediction from axioms (this is very similar to the distinction reviewer 1 makes between description of patterns and analysis of causes). Also, there is not so much mathematical modelling in psychology and social science as evolutionary biology. We thus develop the expectation that in psychological/social science studies, ‘life history theory’ will come less and less to mean formal mathematical models, and more and more to mean a particular empirical pattern to look for. Since the ‘fast-slow continuum’ is the empirical pattern most often mentioned in conjunction with ‘life history theory’, this translates to the prediction: the more the research comes from psychology and social science, the less mathematical modelling there will be, and the more the empirical pattern of the ‘fast-slow’ continuum will come to be the main focus. We hope that this is sufficiently clear now.

The methods include sufficient detail to replicate the study, and the most important research decisions

(i.e. bibliographic coupling over co-citation) are well-justified. I'm also happy to see that the researchers

included the results of different cluster resolutions in the supporting information. I think clarification is

needed in a couple of places, though. For example, why was the default VOSviewer cut-off of 500

documents (see line 124) chosen rather than the largest connected network? Depending on network size in each time period, this decision could skew the measured levels of connectivity.

We chose the 500-document cut off on the basis of published rules of thumb for using VOS Viewer (specifically, p. 19 [preprint pagination] in Van Eck, N.J., & Waltman, L. (2014).

Visualizing bibliometric networks. In Y. Ding, R. Rousseau, & D. Wolfram (Eds.), *Measuring scholarly impact: Methods*

and practice (pp. 285–320). Springer [also available as a preprint from VoS Viewer website].). We have now investigated using the largest connected network instead. The pragmatic reason for limiting to the 500 most-connected documents is that if one does not do this, the map can end up with some extreme outliers a long way from the main body of papers. For example, here is the post-2010 map using the largest connected network (914 documents) instead of the 500 best-connected, leaving all other parameters the same.

dawe (2012)

There are some differences between this and the analysis we presented in the paper, above and beyond the extreme outliers. In particular, in the above map there are just four clusters rather than five: B2 (human developmental and evolutionary psychology) has merged with B4 (human evolutionary anthropology). Thus, no doubt some of the follow-up inferences would be slightly different if one followed this approach. However, there are also a lot of similarities, namely: two non-human clusters; a cluster division of the ‘dark triad’ work from other human work; a largely linear shape with the human work ‘poking out’; the ‘dark triad’ work being furthest from any non-human work and connected only indirectly to it. Thus, we believe that though there are many subtle differences in analytic strategy we could have used, and these would have made some difference, many of our main claims are robust to changes in researcher decisions. (The pre-2010 map and clustering is hardly changed by using the largest connected network instead of just the best-connected 500). Our strategy, though not the only possible one, is reasonable. Therefore, we have: (i) cited the van Eck book chapter as the reason for limiting to the 500 best-connected documents; and (ii) stated in methods that this makes small differences to the cluster detection but produces a similar shape for the core of the maps; (iii) Added a limitations paragraph towards the end of the discussion, where we explicitly acknowledge that there are different parameter settings that could be chosen and that cluster boundaries in particular are rather affected by these choices.

Using only 10 sample papers from each cluster also seems quite low to me. For larger clusters, less of the intellectual diversity will be characterized. This limitation should at least be acknowledged in the discussion.

With 10 clusters in total to cover, downloading and reading 10 per cluster equates to 100 papers. We admit that we are not capturing all the diversity; we see it as more of a

transect through each one to get an idea of the kinds of material most often covered. The new limitations paragraph at the end of the discussion includes the sentence: "Finally, for our assessment of the content of research areas, we have only sampled ten papers from each cluster. Whilst this gives sufficient evidence for describing broad differences between clusters, it will under-estimate the intellectual diversity within each cluster. Each of our clusters could itself be the subject of a detailed qualitative review."

Lines 144-148 are a bit unclear. Were the proportion of possible links calculated for each pair of clusters? Or just within clusters?

We have rephrased here to more explicitly state that the proportion of possible links was calculated for every possibly pairing of clusters (including the pairing of a cluster to itself) within a time period. We also noticed that in this paragraph we tended to slip interchangeably between 'connections' and 'links', and so we have standardized this.

At the beginning of the results, the authors say that after "an initial phase of exponential growth, the number of new papers has grown linearly since around 2005". It seems in the figure that an exponential function would fit the data well, so it might be worth replacing the loess fit with that and changing the wording in the results.

In fact, we have tried fitting an exponential function and it does not really fit very well, despite the wording we used (an exponential function that correctly fits the increase through the 1990s hugely over-predicts the number of papers per year there should be by now). The pattern is actually almost perfectly characterised by no growth in the 1980s, linear growth at one rate from 1990 until about 2004, and then linear growth at a higher rate from 2004 until the present. However, all we meant to say here is that the number of papers per year using the term 'life history theory' had a slow start but then grew steadily. We have rephrased to say just this.

Figure 2 uses the same colors for both maps, which is briefly mentioned in the caption but should be expanded. I think that most readers will mistakenly assume that the clusters are analogous between time periods, so the researchers should take extra care to clarify this.

We have now re-colored all the figures manually so that there is a reasonable degree of follow-on from panel to panel and figure to figure. Research on humans is always shown in green, with different shades of green as required. Bird clusters are always shown in red; mammals in purple; mixed birds and mammals dark red, and so on. In some cases in the Supporting Information figures, this scheme becomes challenging (e.g. because of mixed clusters or clusters that have no direct counterpart in another time period), but we feel that it works reasonably well overall. We have also removed the individual colouring of each bar from figure 3 (and the legend). These did not actually carry any vital information, and the colours did not map onto those of figure 2.

In the discussion (see lines 232-233) the researchers mention that "the three human clusters... lie at progressively greater bibliometric distance from life history theory as it is practiced in ecology and

evolutionary biology". Were the distances between clusters actually calculated, or is this just a qualitative conclusion?

This was really just a verbal description of what we see in figure 2B. We have rephrased to 'The three human clusters—evolutionary anthropology, developmental/personality psychology, and dark triad—appear on the map at successively greater distances from the clusters of non-human papers'.

- Line 51: "become different kinds of thing?"

Not sure what the referee is highlighting here, but we have rephrased to 'are in fact different kinds of thing.'

- Line 104: "of the search term in title, abstract..."

We have added a 'the', hoping that this was what the reviewer was after.

- Line 124: "We created maps were based..."

Thanks, also noted by reviewer 1, corrected.

- Lines 192-194: Sentence could be broken up

Thanks, done.

- Lines 340-342: Sentence could be broken up

Thanks, done.

Appendix B

Response to referees

Thank you for helpful comments on this paper, which we have enjoyed responding to. The main issues you highlighted were, first, analysis of alternate search terms. We have now done this, and report it in the revised section 1 of the supporting information, also drawing attention to it in the discussion, and explaining at the outset why we used the search term 'life history theory' for the main part of the paper (see first response to referee 1). Second, reorganization of the introduction to make the logic and aims clearer. We have rewritten the introduction to consolidate the aims at the end, as suggested (see first response to referee 2).

In addition, we have made a number of more minor changes, as detailed below. For the comments of referee 1, there is a lot more than could be said on the history of life history theory and the validity of some of its extensions to humans. However, in view of the nature of the paper and *Proceedings* word length, we have restricted ourselves to a few brief extra points in the discussion.

Referee: 1

There was a lot of life history theory published before Charnov (1979) that this search is not picking up.

The most significant conceptual seed was Williams (1966a,b), both the book and the paper. A key

publication presenting formal theory was Gadgil & Bossert (1970). Did your search reveal references to

Brian Charlesworth's papers and book? A citation analysis that misses those sources is suspect and

reinforces the need to strengthen the analysis by examining how robust the results are to the

formulation of the search term. Would the results change significantly if the search had been

performed using "life history evolution" or "life history strategy" rather than "life history theory"?

Thanks for these suggestions. To be clear, the object of our study was the particular label 'life history theory', since this has become a very recognisable 'badge' in both biology and psychology/social science. We are principally interested in the evolution of the use of this label, rather than a full bibliometric review of the field of life history evolution. It is clear that searching for the 'life history theory' label does not detect the entire literature on life history evolution, as the referee rightly states. In the early decades, people did not use this exact label. However, we have now also done a larger search: "'life history theory' OR 'life history strategies' OR 'life history evolution'." We report the results of the broader search in the Supporting Information, section 1.

The broader search returns four times as many records as our 'life history theory' search. It captures more of the early work, though still not all of it, which suggests that digitisation

on Web of Science is patchy in the 1970s. It captures a great deal more work on plants and fishes, and the cluster structure and shape of the map are somewhat different from the analyses reported in the main paper (see Supporting Information, section 1). However, the central point remains in this broader analysis: there has been a large growth in human psychological/social work that alludes to life history theory/strategies/evolution; and this work is in its own cluster(s), rather isolated from the non-human literatures.

We have taken the following actions to make our aims clearer and our analysis more robust:

1. Rewritten the introduction to clarify that we aimed to study the evolution of the particular label 'life history theory', even though this exact label was not usually used in the seminal early work, and that we are not claiming to have reviewed all or even a representative sample of all literature on life history evolution.
2. Expanded and modified section 1 of the Supporting Information to cover the broader search described above and present the main results of analysing that search.
3. Alluded to this limitation in the penultimate paragraph of the discussion, briefly mentioning the conclusions that the broader search leads to, and how these differ from the narrower one 'life history theory'.

Line 124: cut "were" – a mistake left from editing.

Thanks, corrected.

Line 138: "taxa from which" not "taxa from whom", "data are" not "data is"

Corrected.

Line 196: One recalls the adage that a work or a concept can become so well known that it is accepted

as part of the general background of a field and no longer needs to be cited because "everybody knows

that". Perhaps you are measuring roughly when this happened to "life history theory": about 2010.

Well, if that is so, it only appears to be so amongst people who work on humans, since Stearns' *Evolution of Life Histories*, and earlier work such as George C. Williams, continues to be highly cited in the clusters B1 and B3 where people are working on non-human species. We suspect, rather, that this is the product of people who work on humans relying on secondary descriptions of what life-history theory is. Therefore, the lack of citation to a common core of work is likely to reflect different people getting their secondary descriptions from different sources. We haven't actually made a change to the paper on this point – exactly what is going on is something of a speculation.

Line 215: Cluster B5 post-2010 appears to be an echo chamber in which fake news reflects off relatively

impermeable walls. See my comment about the personal impact of your analysis at the end of this review.

We might well agree with the reviewer here—at any rate, this is a cluster of work that refers to itself in a relatively insular way. But in this paper we are trying to present the data in an objective way, without any value judgements about any of the work captured by the searches. We will leave readers to make their own inferences about the scientific dynamics in the various clusters.

Line 236: The deep and important distinction between the analysis of causes and the description of patterns is nowhere more evident than in the superficiality suggested by the description of what is going on in post-2010 clusters B2 and B5. One would at least hope that instead of simply invoking a biological substrate for social and psychological patterns that these folks would explicitly consider the possibility that there might be purely cultural processes that could generate all or part of the patterns described. That is, of course, a problem with their papers, not with your analysis, which does a pretty good job of revealing what is going on. You are perfectly correct to raise the issue, what does the word theory mean in that literature? Evidently it is more of an excuse to make one comfortable than an explanation of an important piece of nature. Here let me recall John Maynard Smith's dichotomization of scientists. There are two kinds, he said. Those who want to be right, and those who want to know the truth. If you want to know the truth, you have to be prepared to admit when you are wrong.

This is a really interesting comment. As the referee says, it is a critique of the work we are reviewing more than of our review. We have, however, sharpened up our distinction between the description of patterns and the analysis of causes, because we think it is useful in terms of understanding what is going on in the different parts of the literature. In particular, it seems right to say that biological 'life history theory' research is typically concerned with the analysis of causes, whereas the human research is more typically concerned with the description of patterns. We set up the expectation that this may be the case in the revised introduction:

By theory, Stearns [32] meant the practice of formal mathematical modelling of fitness in relation to life history traits, rather than any particular empirical claim that might arise from such models. However, in psychology and most social sciences, the term 'theory' refers to frameworks that are not formalized, and are at least to some extent inductive (based on typical patterns in data) rather than deductive (based on logical inferences from axioms). If 'life history theory' has adapted to the substrate of psychology and the social sciences as it has spread, we should expect decreasing use of formal models, and an increasingly close association of 'life history theory' with a characteristic empirical pattern. Our impression (from, for example [4]) is that this has

indeed happened. In many psychological and social-science works, 'life history theory' is used as a near-synonym for the 'fast-slow continuum' (or 'fast' and 'slow' strategies). This is the descriptive generalization that variation between species or between individuals can be organized onto a principal axis from early maturation and reproduction, small body size, large numbers of offspring and low parental investment at one end; to late maturation and reproduction, large body size, small numbers of offspring and high parental investment at the other. A full review of the sources, varieties of, and evidence for, the 'fast-slow continuum' idea is beyond our scope here. We merely hypothesise that it might play a different role in 'life history theory' in psychology and social science as compared to ecology and evolution.

We then return to this distinction in the discussion, in the passage on the widespread appeal to the 'fast-slow' continuum idea in psychology and the social sciences:

We suggest that the strong focus on the 'fast-slow' construct represents a shift from seeing 'life history theory' as a kind of methodology (in ecology and evolution), to seeing it as search for a characteristic empirical pattern (in psychology and social science). This would be an accommodation in the understanding of 'life history theory' to what other theories typically look like in those disciplines. Notably, the origins of the 'fast-slow' terminology are inductive, arising from empirical research on cross-species patterns of covariation in multiple life history traits [43,44]. Thus, the fast-slow concept thus arose not from life history theory (in the sense of formal modelling), but as an inductive generalization from comparative data.

Line 280: The controversy about the meaning of r&k-selection in the late 1970's has strong parallels to the contrast you draw between descriptions of slow-fast patterns and theoretical analyses of the evolutionary causes that produce those patterns. Here again the use of particular search terms misses the origin of concepts, which evidently go through a period of incubation before being fossilized in a particular form. The slow-fast continuum, earlier referred to as r&k selection by MacArthur and Wilson (1967) and Pianka (1970) was quantified as Principal Component 1 in Stearns, S.C. 1983. The influence of size and phylogeny on patterns of covariation in the life-history traits of mammals. Oikos 41: 173-87. The fact that the percentage of total variation that it captured dropped from 68-75% to 29-36% when order and family effects were removed tells us a lot about the portion of variation that theory was explaining, for theory did not include the clade-specific effects that one might call phylogenetic

constraint or inertia. The pattern being invoked, in other words, was more complicated and not as strong as those working in areas B2 and B5 appear to believe and had causes that were at least as deeply rooted in the ancient and inherited design of large groups of organisms as in any immediate selection produced by variation in demographic vital rates. George Santayana: "Those who cannot remember the past are condemned to repeat it." (Some think we are condemned to repeat it whether we remember it or not, but I would cite the existence of the EU and UN, however imperfect they may be, as evidence that counts against that claim.)

Thanks for this. This is useful background. In fact, reading what psychologists write when they talk about 'fast-slow', it seems that they have 'r/K' strategists out of Pianka (1970) pretty squarely in mind. It is thus ironic that those models fell out of view so long ago in biology, and yet psychologists are still adopting the idea as if it were biological orthodoxy. Length constrains us from going into this in too much detail here. In the discussion of the 'pace of life' paradigm and human life history theory, we have however added the sentence:

(Indeed, it is somewhat ironic that modern biological life history theory was largely founded on the rejection of a simple 'r-K' continuum and its corresponding causal explanations [27,53], and yet in human research, such a pattern is sometimes taken as definitive of 'life history theory').

Lines 289-290: spot on.

Thank you.

Lines 298-319: See comments on Line 280. In addition, one could point out that one concept that connects life history theory to pace of life discussions is that of adaptive reaction norms. One can conceptually extend the very well analyzed idea of reaction norms for age and size at maturity, now commonplace in the fisheries literature, to reaction surfaces for syndromes of life history traits, which is what pace of life describes.

We have added a brief reference to this point, with a citation to Stearns and Koella (1986) on reaction norms in understanding life history variation.

Lines 333-335: You make very good points about the role that a core body of theory plays in stabilizing a field. It also plays an essential role in checking the logic of claims and reducing the possibility of error introduced by the kind of story-telling that aims at confirming pet hypotheses rather than demanding

that claims about reality stand up to strong confrontation with alternative hypotheses. Consider adding a comment to that effect.

This strikes us a potentially complex issue. There are some areas (cultural evolution theory springs to mind) where having a body of core formal theory has led people to look for the phenomena assumed or predicted in the formal models, rather than really describing how cultural phenomena actually work. So the point about confirmation bias appears to us to be orthogonal to the issue of having formal core theory or not.

I found myself quite emotionally engaged with this paper because it recalls issues that both shaped and threatened my career in science. When I abandoned r&K selection as an organizing principle after having used it in my 1976 review paper, which was based on my 1973 PhD proposal, and pointed out, along with Brian Charlesworth (who was ahead of me on this), that the analysis of the evolutionary consequences of age-specific variation in birth and death rates got us a lot closer to causes than did coarse-grained analogies to the Lotka-Volterra model of density dependent population regulation, I made an enemy of E.O. Wilson for life. Fortunately, he was not the only influential figure in the field, and George Williams became a supporter and a friend.....I would also like to register that I am deeply offended at the way that some evolutionary psychologists have used references to life history theory to support claims that pace of life and slow-fast analogies explain variation in tendencies towards criminality. Intellectual dishonesty often contributes to moral failure. See, for example, Peter Rosenthal's paper, "The criminality of racial harassment," (1989-1990 Can. Hum. Rts. Y.B. 113 (1989-1990) which comments on Phillippe Rushton's use of r&K selection to support racist propaganda.

Thanks for these comments, especially on your view of some of the human research. We are now working on a more qualitative paper on 'life history theory' as it is understood in the human sciences. In that paper, we would love to explore the issues a bit more critically. In this paper, we are trying to stay focussed on the objective bibliometrics, and so we have not made any major change in response to the above. We have added to the discussion the following brief point:

Clusters B2 and B5 extend the fast-slow continuum idea....from between-species variation to within-species variation; from biometric to behavioural and personality variables; and (in B2 especially) from genetic evolution to individual plasticity based on developmental experience. The validity of those extensions is beyond our scope here, but their theoretical justification is not trivial.

Referee: 2

The introduction provides sufficient background on both life history theory and bibliometric methods, but the aims and hypotheses are scattered in a way that is hard to follow. The first aim appears in lines 51-54, followed by two paragraph on bibliometrics, followed by

another aim on lines 85-86, and then a hypothesis at the end of the last paragraph. This should be restructured so that the aims and hypotheses appear in the same location.

We have restructured the introduction so that the aims are consolidated at the end, instead of being peppered throughout. To facilitate this, we have moved one paragraph (a relatively detailed one on how bibliometric analysis is done) out of the introduction to the first paragraph of the methods. We hope that the aims and hypotheses are clearer now.

Additionally, the logic of the last hypothesis in the last paragraph of the intro seems weak compared to the justification in the discussion. Try to make the distinction between the hypothesized differences between biology and the social sciences more explicit.

We have tried to flesh out the logic of our hypothesis without pre-empting the findings by expanding our discussion of what ‘theories’ typically look like in psychology and social science (see penultimate paragraph of revised introduction). A typical psychological theory is based on identifying a *characteristic pattern* rather than a priori prediction from axioms (this is very similar to the distinction reviewer 1 makes between description of patterns and analysis of causes). Also, there is not so much mathematical modelling in psychology and social science as evolutionary biology. We thus develop the expectation that in psychological/social science studies, ‘life history theory’ will come less and less to mean formal mathematical models, and more and more to mean a particular empirical pattern to look for. Since the ‘fast-slow continuum’ is the empirical pattern most often mentioned in conjunction with ‘life history theory’, this translates to the prediction: the more the research comes from psychology and social science, the less mathematical modelling there will be, and the more the empirical pattern of the ‘fast-slow’ continuum will come to be the main focus. We hope that this is sufficiently clear now.

The methods include sufficient detail to replicate the study, and the most important research decisions

(i.e. bibliographic coupling over co-citation) are well-justified. I'm also happy to see that the researchers

included the results of different cluster resolutions in the supporting information. I think clarification is

needed in a couple of places, though. For example, why was the default VOSviewer cut-off of 500

documents (see line 124) chosen rather than the largest connected network? Depending on network size in each time period, this decision could skew the measured levels of connectivity.

We chose the 500-document cut off on the basis of published rules of thumb for using VOS Viewer (specifically, p. 19 [preprint pagination] in Van Eck, N.J., & Waltman, L. (2014).

Visualizing bibliometric networks. In Y. Ding, R. Rousseau, & D. Wolfram (Eds.), *Measuring scholarly impact: Methods*

and practice (pp. 285–320). Springer [also available as a preprint from VoS Viewer website].). We have now investigated using the largest connected network instead. The pragmatic reason for limiting to the 500 most-connected documents is that if one does not do this, the map can end up with some extreme outliers a long way from the main body of papers. For example, here is the post-2010 map using the largest connected network (914 documents) instead of the 500 best-connected, leaving all other parameters the same.

dawe (2012)

There are some differences between this and the analysis we presented in the paper, above and beyond the extreme outliers. In particular, in the above map there are just four clusters rather than five: B2 (human developmental and evolutionary psychology) has merged with B4 (human evolutionary anthropology). Thus, no doubt some of the follow-up inferences would be slightly different if one followed this approach. However, there are also a lot of similarities, namely: two non-human clusters; a cluster division of the ‘dark triad’ work from other human work; a largely linear shape with the human work ‘poking out’; the ‘dark triad’ work being furthest from any non-human work and connected only indirectly to it. Thus, we believe that though there are many subtle differences in analytic strategy we could have used, and these would have made some difference, many of our main claims are robust to changes in researcher decisions. (The pre-2010 map and clustering is hardly changed by using the largest connected network instead of just the best-connected 500). Our strategy, though not the only possible one, is reasonable. Therefore, we have: (i) cited the van Eck book chapter as the reason for limiting to the 500 best-connected documents; and (ii) stated in methods that this makes small differences to the cluster detection but produces a similar shape for the core of the maps; (iii) Added a limitations paragraph towards the end of the discussion, where we explicitly acknowledge that there are different parameter settings that could be chosen and that cluster boundaries in particular are rather affected by these choices.

Using only 10 sample papers from each cluster also seems quite low to me. For larger clusters, less of the intellectual diversity will be characterized. This limitation should at least be acknowledged in the discussion.

With 10 clusters in total to cover, downloading and reading 10 per cluster equates to 100 papers. We admit that we are not capturing all the diversity; we see it as more of a

transect through each one to get an idea of the kinds of material most often covered. The new limitations paragraph at the end of the discussion includes the sentence: "Finally, for our assessment of the content of research areas, we have only sampled ten papers from each cluster. Whilst this gives sufficient evidence for describing broad differences between clusters, it will under-estimate the intellectual diversity within each cluster. Each of our clusters could itself be the subject of a detailed qualitative review."

Lines 144-148 are a bit unclear. Were the proportion of possible links calculated for each pair of clusters? Or just within clusters?

We have rephrased here to more explicitly state that the proportion of possible links was calculated for every possibly pairing of clusters (including the pairing of a cluster to itself) within a time period. We also noticed that in this paragraph we tended to slip interchangeably between 'connections' and 'links', and so we have standardized this.

At the beginning of the results, the authors say that after "an initial phase of exponential growth, the number of new papers has grown linearly since around 2005". It seems in the figure that an exponential function would fit the data well, so it might be worth replacing the loess fit with that and changing the wording in the results.

In fact, we have tried fitting an exponential function and it does not really fit very well, despite the wording we used (an exponential function that correctly fits the increase through the 1990s hugely over-predicts the number of papers per year there should be by now). The pattern is actually almost perfectly characterised by no growth in the 1980s, linear growth at one rate from 1990 until about 2004, and then linear growth at a higher rate from 2004 until the present. However, all we meant to say here is that the number of papers per year using the term 'life history theory' had a slow start but then grew steadily. We have rephrased to say just this.

Figure 2 uses the same colors for both maps, which is briefly mentioned in the caption but should be expanded. I think that most readers will mistakenly assume that the clusters are analogous between time periods, so the researchers should take extra care to clarify this.

We have now re-colored all the figures manually so that there is a reasonable degree of follow-on from panel to panel and figure to figure. Research on humans is always shown in green, with different shades of green as required. Bird clusters are always shown in red; mammals in purple; mixed birds and mammals dark red, and so on. In some cases in the Supporting Information figures, this scheme becomes challenging (e.g. because of mixed clusters or clusters that have no direct counterpart in another time period), but we feel that it works reasonably well overall. We have also removed the individual colouring of each bar from figure 3 (and the legend). These did not actually carry any vital information, and the colours did not map onto those of figure 2.

In the discussion (see lines 232-233) the researchers mention that "the three human clusters... lie at progressively greater bibliometric distance from life history theory as it is practiced in ecology and

evolutionary biology". Were the distances between clusters actually calculated, or is this just a qualitative conclusion?

This was really just a verbal description of what we see in figure 2B. We have rephrased to 'The three human clusters—evolutionary anthropology, developmental/personality psychology, and dark triad—appear on the map at successively greater distances from the clusters of non-human papers'.

- Line 51: "become different kinds of thing?"

Not sure what the referee is highlighting here, but we have rephrased to 'are in fact different kinds of thing.'

- Line 104: "of the search term in title, abstract..."

We have added a 'the', hoping that this was what the reviewer was after.

- Line 124: "We created maps were based..."

Thanks, also noted by reviewer 1, corrected.

- Lines 192-194: Sentence could be broken up

Thanks, done.

- Lines 340-342: Sentence could be broken up

Thanks, done.